# Neural Sparse Representation for Image Restoration

**Yuchen Fan, Jiahui Yu, Yiqun Mei, Yulun Zhang$^{†}$, Yun Fu$^{†}$, Ding Liu$^{‡}$, Thomas S. Huang**
University of Illinois at Urbana-Champaign, $^{†}$Northeastern University, $^{‡}$ByteDance
yuchenf4@illinois.edu

## Abstract

Inspired by the robustness and efficiency of sparse representation in sparse coding based image restoration models, we investigate the sparsity of neurons in deep networks. Our method structurally enforces sparsity constraints upon hidden neurons. The sparsity constraints are favorable for gradient-based learning algorithms and attachable to convolution layers in various networks. Sparsity in neurons enables computation saving by only operating on non-zero components without hurting accuracy. Meanwhile, our method can magnify representation dimensionality and model capacity with negligible additional computation cost. Experiments show that sparse representation is crucial in deep neural networks for multiple image restoration tasks, including image super-resolution, image denoising, and image compression artifacts removal. Code is available at `https://github.com/ychfan/nsr`.

## 1   Introduction

Sparse representation plays a critical role in image restoration problems, such as image super-resolution [1, 2, 3], denoising [4], compression artifacts removal [5], and many others [6, 7]. These tasks are inherently ill-posed, where the input signal usually has insufficient information while the output has infinitely many solutions w.r.t. the same input. Thus, it is commonly believed that sparse representation is more robust to handle the considerable diversity of solutions.

Sparse representation in sparse coding is typically *high-dimensional* but with *limited non-zero* components. Input signals are represented as sparse linear combinations of tokens from a dictionary. High dimensionality implies larger dictionary size and generally leads to better restoration accuracy, since a more massive dictionary is capable of more thoroughly sampling the underlying signal space, and thus more precisely representing any query signal. Besides, sparsity limits numbers of non-zero elements work as an essential image prior, which has been extensively investigated and exploited to make restoration robust. Sparsity also brings computational efficiency by ignoring zero parts.

Deep convolutional neural networks for image restoration [8, 9, 10, 11, 12] extend the sparse coding based methods with repeatedly cascaded structures. The deep network based approach was firstly introduced to improve the performance in [13] and conceptually connected with previous sparse coding based methods. A simple network, with two convolutional layers bridged by a non-linear activation layer, can be interpreted as: activation denotes sparse representation; non-linearity enforces sparsity and convolutional kernels consist of the dictionary. SRResNet [14] extended the basic structure with skip connection to form a residual block and cascaded a large number of blocks to construct very deep residual networks.

Sparsity of hidden representation in deep neural networks cannot be efficiently solved by iterative optimization as sparse coding, since deep networks are feed-forward during inference. Sparsity of neurons is commonly achieved by ReLU activation in [15] by thresholding negative values to zero independently in each neuron. Still, its 50% sparsity on random vectors is far from the sparsity definition on the overall number of non-zero components. Oppositely, sparsity constraints are more

actively used in model parameters to achieve network pruning [16]. However, hidden representation dimensionality is reduced in pruned networks, and accuracy may hurt.

In this paper, we propose a method that can structurally enforce sparsity constraints upon hidden neurons in deep networks but also keep representation in high dimensionality. Given high-dimensional neurons, we divide them into groups along channels and allow only one group of neurons can be non-zero each time. The adaptive selection of the non-sparse group is modeled by tiny side networks upon context features. And computation is also saved when only performed on the non-zero group. However, the selecting operation is not differentiable, so it is difficult to embed the side networks for joint training. We relax the sparse constraints to soft and approximately reduce as a sparse linear combination of multiple convolution kernels instead of hard selection. We further introduce additional cardinal dimensions to decompose sparsity prediction into sub-problems by splitting each sparse group and concatenating after cardinal-independent combination of parameters.

To demonstrate the significance of neural sparse representation, we conduct extensive experiments on image restoration tasks, including image super-resolution, denoising, and compression artifacts removal. Our experiments conclude that: (1) dedicated constraints are essential to achieve neural sparsity representation and benefit deep networks; (2) our method can significantly reduce computation cost and improve accuracy, given the same size of model footprint; (3) our method can dramatically enlarge the model capacity and boost accuracy with negligible additional computation cost.

## 2    Related Work

### 2.1    Sparse coding and convolutional networks

Here we briefly review the application of sparsity in image restoration and its relation to convolutional networks. Considering image super-resolution as an example of image restoration, sparse coding based method [1] assumes that input image signal $X$ can be represented by a sparse linear combination $\alpha$ over dictionary $D_1$, which typically is learned from training images as

$$X \approx D_1\alpha, \text{ for some } \alpha \in \mathbb{R}^n \text{ and } \|\alpha\|_0 \ll n. \tag{1}$$

In [3], a coupled dictionary, $D_2$, for restored image signal $Y$ is jointly learned with $D_1$ as well as its sparse representation $\alpha$ by

$$Y \approx D_2\alpha. \tag{2}$$

Convolutional networks, which consist of stacked convolutional layers and non-linear activation functions, can be interpreted with the concepts from sparse coding [13]. Given for instance a small piece of network with two convolutional layers with kernels $W_1, W_2$ and a non-linear function $F$, the image restoration process can be formalized as

$$Y = W_2 * F(W_1 * X). \tag{3}$$

The convolution operation $*$ with $W_1$ is equivalent to projecting input image signal $X$ onto dictionary $D_1$. The convolution operation $*$ with $W_2$ is corresponding to the projection of the signal representation on dictionary $D_2$. These two convolutional layers structure is widely used as a basic residual block and stacked with multiple blocks to form very deep residual networks in recent advances [14, 8] of image restoration.

Dimensionality of hidden representation or number of kernels in each convolutional layer determines the size of dictionary memory and learning capacity of models. However, unlike sparse coding, representation dimensionality in deep models is usually restricted by running speed or memory usage.

### 2.2    Sparsity in parameters and pruning

Exploring the sparsity of model parameters can potentially improve robustness [17], but sparsity in parameters is not sufficient and necessary to result in sparse representation. Furthermore, group sparsity upon channels and suppression of parameters close to zero can achieve node pruning [18, 16, 19, 20, 21], which dramatically reduces inference computation cost. Despite efficiency, node pruning reduces representation dimensionality proportionally instead of sparsity, limits representation diversity, and leads to accuracy regression.

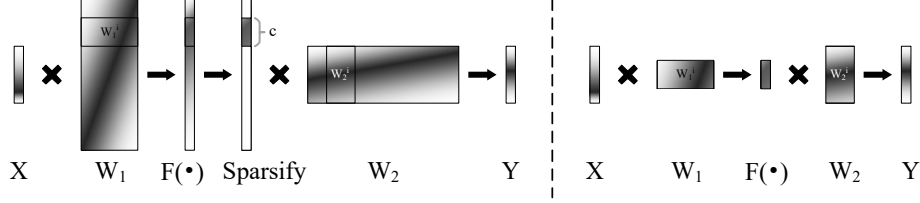

$$X \qquad W_1 \qquad F(\bullet) \quad \text{Sparsify} \qquad W_2 \qquad\qquad Y \qquad\quad X \qquad W_1 \qquad F(\bullet) \qquad W_2 \qquad Y$$

Figure 1: Illustration of computation reduction in two-layer neural networks with sparse hidden nodes in a simplified matrix multiplication example. Left: network with sparsity constraints, which only allow one group with $c$ hidden nodes to be non-zero over $kc$ nodes in total. Right: reduced computation with only $W_1^i$ and $W_2^j$, since other activation nodes are zero. (Grayscale reflects magnitude of matrix values. Matrix multiplication is in right to left order.)

## 2.3 Thresholding and gating

Thresholding function, ReLU [22] for example, plays the similar role of imposing the sparsity constraints [15] by filtering out negative values to zero, and contributes to significant performing improvement over previous activation functions, *i.e.*, hyperbolic tangent. Although ReLU statistically gives only 50% sparsity over random vectors, there is still a significant gap between sparsity definition in Eq. 1. Gating mechanism, in Squeeze-and-Excitation [23, 24], for example, scales hidden neurons with adaptive sigmoid gates and slightly improves sparsity besides noticeable accuracy improvements. Both thresholding and gating are applied independently to hidden neurons and could not inherently guarantee global sparsity in Eq. 1.

# 3 Methodology

We propose novel sparsity constraints to achieve sparse representation in deep neural networks. Relaxed soft restrictions are more friendly to gradient-based training. Additional cardinal dimension refines the constraints and improves the diversity of sparse representation.

## 3.1 Sparsity in hidden neurons

Unlike the methods discussed in Section 2.3 only considering local sparsity for each neuron independently, our approach enforces global sparsity between groups. Specifically, the hidden neurons are divided into $k$ groups with $c$ nodes in each group, and only one group is allowed to contain non-zero values. Correspondingly, convolution kernels can also be divided upon connected hidden neurons. Then only the kernels connected to non-zero neurons need to be accounted. Formally, for networks structure in Eq. 3, the convolution kernels are divided as $W_1 = [(W_1^1)^T, (W_1^2)^T, \dots (W_1^k)^T]^T$ and $W_2 = [W_2^1, W_2^2, \dots W_2^k]$. Then the Eq. 3 can be rewritten as

$$
\begin{aligned}
Y &= [W_2^1, W_2^2, \dots W_2^k] F([(W_1^1)^T, (W_1^2)^T, \dots (W_1^k)^T]^T X) \\
&= \sum_{i=1}^{k} W_2^i * F(W_1^i * X).
\end{aligned}
\tag{4}
$$

When sparsity constraints only allow the $i$th group of neurons with non-zero components, then Eq. 4 can be reduced, as shown in Figure 1, and formally as

$$Y = W_2^i * F(W_1^i * X). \tag{5}$$

The proposed sparsity is supposed to pick the node group with the largest amplitude and cannot be achieved without computing the values of all the nodes. In our approach, the selection of the only non-zero group is modeled by a multi-layer perceptron (MLP) with respect to the input signal $X$.

Regular convolution operations need the kernels shared for every pixel. Hence the selection should also be identified through the spatial space. We are inspired by the Squeeze-and-Excitation [23, 24] operation and propose to add pooling operation before the MLP and boardcasting operation for group selection. The above procedure can be formalized as

$$i = \operatorname*{argmax}_{j \in [\![1,k]\!]} \operatorname{MLP}(\operatorname{Pool}(X), j). \tag{6}$$

Note that, as most of patch-based algorithms [1, 24] for image restoration, the pooling operation should be with respect to a specific patch size instead of the whole image.

**Comparison to thresholding and gating.** The proposed method limits the number of non-zero entities under $1/k$ of all the nodes in hidden-layer representation, which is more closed to sparsity definition in Eq. 1 than thresholding and gating methods discussed in section 2.3. The proposed method also dramatically reduces computation cost by $k$ times by only considering the adaptively selected group, which is not possible with thresholding and gating methods.

**Comparison to node pruning.** Node pruning is designed to diminish activation nodes by zeroing all the related trainable parameters. The pruned nodes stick to zero no matter how the input signal varies, which substantially reduces representation dimensionality. In our method, the sparsity adaptively depends on input. Although the input inherently keeps the high dimensionality in representation, our method saves computation and memory cost as narrow models.

**Comparison to block sparse coding.** Our method with multi-level sparsity constraints, by ReLU activation in the main networks and Softmax activation in the side MLP, shares a similar idea with block sparsity [25]. The sparsity constraints are enforced into model structure instead of additional objective functions which is more favorable for efficiency and optimization in deep models.

**Comparison to group sparse coding.** Group sparse coding [26] divides input instances into groups and ensures robust and stable sparse patterns within groups. Our method shares the sparsity group selection for every pixel in an image patch, which achieves robustness and efficiency simultaneously. Experiments show that appropriate patch size benefits model accuracy.

### 3.2 Relaxed soft sparsity

Similar as L0-norm in sparse coding, the adaptive sparse group selection in Eq. 6 is not differentiable and feasible to be jointly learned with neural networks. Although Gumbel trick [27] is proposed to re-parameterize the $\mathrm{argmax}$ with respect to a conditional probability distribution, it does not achieve convincing results in our experiment settings.

The sparsity constrains are relaxed by substituting selection with softmax as a smooth approximation of max. Instead of predicting index over $k$, the MLP is relaxed to predict probability over groups $\beta = [\beta_1, \beta_2, \ldots \beta_k] \in \mathbb{R}^k_{(0,1)}$ with softmax function $\sigma(\cdot)$ by

$$\beta = \sigma(\mathrm{MLP}(\mathrm{Pool}(X))). \tag{7}$$

Then, the two-layer structure in Eq. 4 is updated to adaptive weighted sum of groups as

$$Y = \sum_{i=1}^{k} \beta_i W_2^i * F(W_1^i * X). \tag{8}$$

With weighted summation, Eq. 8 cannot be directly reduced as Eq.5, since none of group weights is exactly zero. Fortunately, given sparse assumption of softmax outputs, $\exists i$, s.t. $\beta_i \gg \beta_j \to 0, \forall j \neq i$, and piece-wise linear activation function $F$, ReLU for example, it can be proved that weighted sum of hidden neurons can be approximately reduced to weighted sum of parameters $W^i$, as shown in Figure 2, and formally as

$$Y \approx \left(\sum_{i=1}^{k} \sqrt{\beta_i} W_2^i\right) * F\left(\left(\sum_{i=1}^{k} \sqrt{\beta_i} W_1^i\right) * X\right). \tag{9}$$

Note that the two $\sqrt{\beta}$ applied to $W_1$ and $W_2$ are not necessary to be identical to achieve the approximation. Our experiments show that independently predicting weights for $W_1$ and $W_2$ has benefits for accuracy.

In this way, networks restricted by soft sparse constraints can be as efficient as those with hard constraints. And the only additional computation cost from the interpolation of convolution kernels is negligible comparing with convolution operations with the image.

**Comparison to conditional convolution.** CondConv [28] has similar operation of the adaptive weighed sum of convolution kernels as our relaxed soft sparsity approach. However, CondConv uses the sigmoid function to normalize the weights of kernels instead of softmax function in our method.

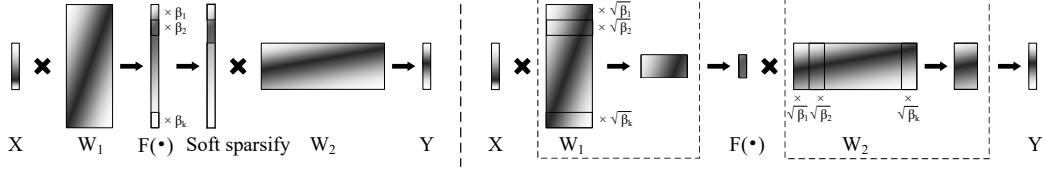

X     $W_1$     F(•)   Soft sparsify   $W_2$     Y      X     $W_1$     F(•)     $W_2$     Y

Figure 2: Illustration of weighted neurons in soft sparsity constraints and reduced counterpart with weighted sum of parameters. Left: network with soft sparsity constraints, weights $\beta_i$ are applied to neurons in $k$ groups. Right: approximate reduction by firstly weighted summing of parameter groups into a small slice then applying it to features.

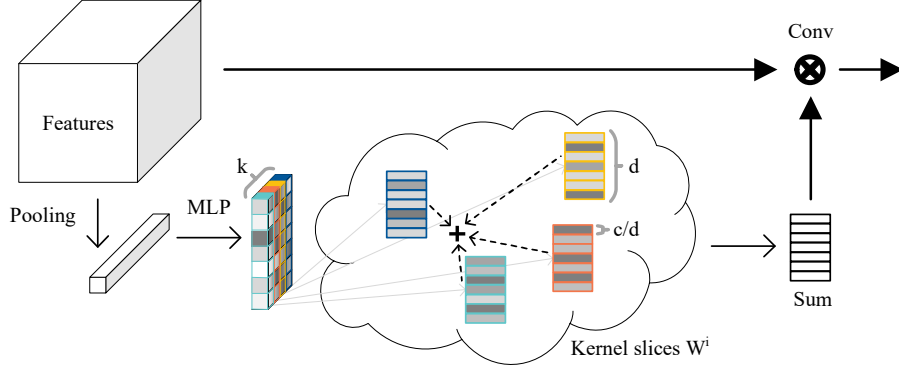

Figure 3: Illustration of our method. Features of image patch are firstly spatially pooled and feed in MLP with softmax activation to predict sparsity constraints $\gamma \in \mathbb{R}^{d,k}$. Softmax function is performed along $k$ axis. Convolution kernel $W$ is divided into $k$ sparsity groups and $c$ channels per group $W^i$. Each group is further divided into $d$ cardinal groups and $c/d$ channels per group $W^{j,i}$. The cardinal-independent weighted sum is performed as Eq. 10. Finally, the aggregated kernel $\hat{W}$ convolves with the original features. (Colors reflect sparsity groups and grayscale reflects magnitude of matrix values.)

Hence, no sparsity constraints are explicitly applied in CondConv, and our experiments show that sparsity is very important for model accuracy. Dynamic convolution [29] also introduces softmax function upon CondConv, but our sparsity weights are conditional on neighbor pixels instead of the whole image which can handle more varieties for image restoration tasks.

### 3.3 Cardinality over sparsity groups

Modeling sparsity between groups with a simple MLP is challenging, especially when dimensionality $c$ per group grows. Also, bonding channels within pre-defined groups limits diversity of the sparsity patterns. Inspired by group convolution in ResNeXt [30], we split the $c$ nodes per sparsity group into $d$ cardinal groups, and each cardinal group with $c/d$ nodes is independently constrained along $k$ sparsity groups, as shown in Figure 3. Formally, the averaging weights are extended to matrix $\gamma = [\gamma_1, \gamma_2, \ldots \gamma_d] \in \mathbb{R}^{d,k}_{(0,1)}$ and $\gamma_i = \sigma(\text{MLP}_i(\text{Pool}(X)))$, then weighted averaged convolution kernel becomes

$$\hat{W} = \operatorname*{concat}_{j=1}^{d} \left( \sum_{i=1}^{k} \gamma_{j,i} W^{j,i} \right),  \tag{10}$$

where $W^i = [W^{1,i}, W^{2,i}, \ldots W^{d,i}]$ and $W^{j,i}$ is the $j$th cardinal group and $i$th sparsity group. concat is concatenation operation along the axis of output channels. Notably, with cardinal grouping, Squeeze-and-Excitation [23] operation becomes a particular case of our approach when $d = c$, $k = 1$ and the MLP activation is substituted with sigmoid function.

# 4 Experiments

## 4.1 Settings

**Datasets and benchmarks.** We use multiple datasets for image super-resolution, denoising, and compression artifacts removal separately. For image super-resolution, models are trained with DIV2K [31] dataset which contains 800 high-quality (2K resolution) images. The DIV2K also comes with 100 validation images, which are used for ablation study. The datasets for benchmark evaluation include Set5 [32], Set14 [2], BSD100 [33] and Urban100 [34] with three up-scaling factors: ×2, ×3 and ×4. For image denoising, training set consists of Berkeley Segmentation Dataset (BSD) [33] 200 images from training split and 200 images from testing split, as [35]. The datasets for benchmark evaluation include Set12, BSD64 [33] and Urban100 [34] with additive white Gaussian noise (AWGN) of level 15, 25, 50. For compression artifacts removal, training set consists of 91 images in [1] and 200 training images in [33]. The datasets for benchmark evaluation include LIVE1 [36] and Classic5 with JPEG compression quality 10, 20, 30 and 40. Evaluation metrics include PSNR and SSIM [37] for predicted image quality in luminance or grayscale, only DIV2K is evaluated in RGB channels. FLOPs per pixels is used to measure efficiency, because the runtime complexity is proportional input image size for fully convolutional models.

**Training settings.** Models are trained with nature images and their degraded counterparts. Online data augmentation includes random flipping and rotation during training. Training is based on randomly sampled image patches for 100 times per image and epoch. And total training epochs are 30. Models are optimized with L1 distance and ADAM optimizer. Initial learning rate is 0.001 and multiplied by 0.2 at 20 and 25 epochs.

## 4.2 Ablation study

We conduct ablation study to prove the significance of neural sparse representation. The experiments are evaluated on DIV2K validation set for image super-resolution with ×2 up-scaling under PSNR. We use WDSR [38] networks with 16 residual blocks, 32 neurons and 4× width multiplier as the baseline, and set $k = 4$ for sparsity groups by default.

**Sparsity constraints.** Sparsity constraints are essential for representation sparsity. We implement the hard sparsity constraints with Gumbel-softmax to simulate the gradient of hardmax and compare it with soft sparsity achieved by softmax function. The temperature in softmax also controls the sharpness of output distribution. When the temperature is small, softmax outputs are sharper and closer to hardmax. Thus gradient will vanish. When the temperature is large, softmax outputs are more smooth, then it will contradict with our sparsity assumption in Eq. 9 for approximation. We also compare them with a similar model with sigmoid function as MLP activation instead of sparsity constraints in CondConv [28]. Results in Table 1 show that Gumbel-based hard-sparsity methods are not feasible and even worse than the baseline without sparsity groups. Temperature is necessary to be initialized with proper value to achieve better results, which coincides with the above analysis. Sigmoid also gets worse results than softmax because sigmoid cannot guarantee sparsity, which also agrees with our comparison in the previous section. In addition, sharing the group selection weights $\beta$ between two convolution layers adjacent to the activation layer drops PSNR from 34.87 dB to 34.85 dB.

**Cardinality.** Cardinal dimension reduces the actual dimensionality and dependency between channels in sparsity groups and improves the diversity of linear combination weights over convolution kernels. Results of models with different cardinalities in Fig. 4 show that increasing cardinality constantly benefits accuracy. We also compare with Squeeze-and-Excitation (SE) model, which is a special case of our method, under the same FLOPs. And our models significantly outperform the SE model.

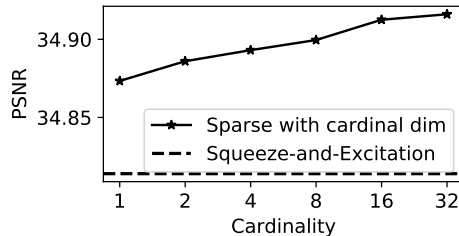

Figure 4: Comparison of cardinality.

**Efficiency.** Our method can approximately save computation by $k$ times with $k$ sparsity groups but remains the same model size or number of parameters. Results in Table 2 have the same model size

Table 1: Comparison of different sparsity constrains.

| Sparsity | N/A | Sigmoid | Gumbel | Softmax($\tau = 10$) | Softmax($\tau = 1$) | Softmax($\tau = 0.1$) |
|---|---|---|---|---|---|---|
| PSNR | 34.76 | 34.81 | 34.45 | 34.86 | **34.87** | 34.83 |

Table 2: Models with the same size. FLOPs is inversely proportional to group size.

| Group size | # of residual blocks | | | |
|---|---|---|---|---|
| | 2 | 4 | 8 | 16 |
| N/A | 33.91 | 34.29 | 34.56 | 34.76 |
| 2 | **33.92** | **34.30** | **34.57** | **34.77** |
| 3 | 33.86 | 34.23 | 34.51 | 34.71 |
| 4 | 33.81 | 34.21 | 34.41 | 34.68 |
| # params (M) | 0.15 | 0.30 | 0.60 | 1.2 |

Table 3: Models with the same FLOPs. Model size is proportional to group size.

| Group size | # of residual blocks | | | |
|---|---|---|---|---|
| | 2 | 4 | 8 | 16 |
| N/A | 33.91 | 34.29 | 34.56 | 34.76 |
| 2 | 33.98 | 34.38 | 34.65 | 34.83 |
| 4 | 34.07 | 34.45 | 34.70 | 34.87 |
| 8 | 34.14 | 34.50 | 34.74 | 34.89 |
| 16 | **34.17** | **34.56** | **34.77** | **34.91** |
| FLOPs (M) | 0.15 | 0.30 | 0.60 | 1.2 |

in columns and show that our method can save at least half of computation without hurting accuracy uniformly for various model sizes.

**Capacity.** Our method can also extend model capacity or number of parameters by $k$ times with $k$ sparsity groups but only with negligible additional computation cost. Results in Table 3 have the same computation cost in columns and show that our method can continually improve accuracy by extending model capacity up to 16 times.

**Computation time.** We follow the protocol as Squeeze-and-Excitation networks using CPU inference time to address the latency for embedded device applications. We evaluate the forward running time of a single convolution layer of our implementation in PyTorch.

Figure 5: Computation time (in milliseconds).

| Group size | N/A | 2 | 4 | 8 | 16 |
|---|---|---|---|---|---|
| fixed model size | 1.96 | 1.17 | 0.77 | 0.56 | 0.70 |
| fixed FLOPs | 1.96 | 2.12 | 2.12 | 2.08 | 2.08 |

The running time is the average of 100 times on quad-core Intel Core i5-2500 at 3.30GHz. We set batch size as 100, input and output channels as 64, patch size as $64 \times 64$. The time for additional computation is around 5-10%.

**Visualization of kernel selection.** It is difficult to directly visualize the sparsity of high-dimensional hidden representation, we take the selection of kernels as a surrogate. As Fig. 6, in the first block, the weights are almost binary everywhere and only depend on color and low-level cues. In later blocks, the weights are more smooth and more attentive to high-frequency positions with more complicated texture. And the last layer is more correlated to high-level context, for example, tree branches in the first image and lion in the second image.

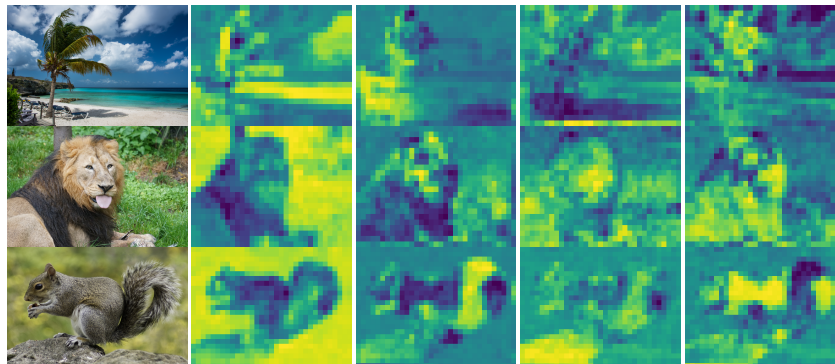

Figure 6: Visualization of kernel selection for network with 2 sparsity group and 4 residual blocks. First column shows input images. Later columns shows softmax MLP outputs of the second convolutional layer in each block. Blue and yellow denotes two groups respectively.

Table 4: Public image super-resolution benchmark results and DIV2K validation results in PSNR / SSIM. The better resutls with small and **large** EDSR are underlined and in bold respectively.

| Dataset | Scale | Bicubic | VDSR | EDSR (S) | Sparse + | EDSR (L) | Sparse - |
|---|---|---|---|---|---|---|---|
| Set5 | ×2 | 33.66 / 0.9299 | 37.53 / 0.9587 | 37.99 / 0.9604 | 38.02 / 0.9610 | 38.11 / 0.9601 | **38.23** / **0.9614** |
| | ×3 | 30.39 / 0.8682 | 33.66 / 0.9213 | 34.37 / 0.9270 | 34.43 / 0.9277 | **34.65** / 0.9282 | 34.62 / **0.9289** |
| | ×4 | 28.42 / 0.8104 | 31.35 / 0.8838 | 32.09 / 0.8938 | 32.25 / 0.8957 | 32.46 / 0.8968 | **32.55** / **0.8987** |
| Set14 | ×2 | 30.24 / 0.8688 | 33.03 / 0.9124 | 33.57 / 0.9175 | 33.60 / 0.9191 | 33.92 / 0.9195 | **33.94** / **0.9203** |
| | ×3 | 27.55 / 0.7742 | 29.77 / 0.8314 | 30.28 / 0.8418 | 30.37 / 0.8443 | 30.52 / 0.8462 | **30.57** / **0.8475** |
| | ×4 | 26.00 / 0.7027 | 28.01 / 0.7674 | 28.58 / 0.7813 | 28.66 / 0.7836 | **28.80** / 0.7876 | 28.79 / 0.7876 |
| B100 | ×2 | 29.56 / 0.8431 | 31.90 / 0.8960 | 32.16 / 0.8994 | 32.26 / 0.9008 | 32.32 / 0.9013 | **32.34** / **0.9020** |
| | ×3 | 27.21 / 0.7385 | 28.82 / 0.7976 | 29.09 / 0.8052 | 29.15 / 0.8074 | 29.25 / 0.8093 | **29.26** / **0.8100** |
| | ×4 | 25.96 / 0.6675 | 27.29 / 0.7251 | 27.57 / 0.7357 | 27.61 / 0.7372 | 27.71 / **0.7420** | **27.72** / 0.7414 |
| Urban100 | ×2 | 26.88 / 0.8403 | 30.76 / 0.9140 | 31.98 / 0.9272 | 32.57 / 0.9329 | 32.93 / 0.9351 | **33.02** / **0.9367** |
| | ×3 | 24.46 / 0.7349 | 27.14 / 0.8279 | 28.15 / 0.8527 | 28.43 / 0.8587 | 28.80 / 0.8653 | **28.83** / **0.8663** |
| | ×4 | 23.14 / 0.6577 | 25.18 / 0.7524 | 26.04 / 0.7849 | 26.24 / 0.7919 | **26.64** / **0.8033** | 26.61 / 0.8025 |
| Manga109 | ×2 | 30.80 / 0.9339 | 37.22 / 0.9750 | 38.55 / 0.9769 | 38.94 / 0.9776 | 39.10 / 0.9773 | **39.31** / **0.9782** |
| | ×3 | 26.95 / 0.8556 | 32.01 / 0.9340 | 33.45 / 0.9439 | 33.77 / 0.9462 | 34.17 / 0.9476 | **34.27** / **0.9484** |
| | ×4 | 24.89 / 0.7866 | 28.83 / 0.8870 | 30.35 / 0.9067 | 30.63 / 0.9106 | 31.02 / **0.9148** | **31.10** / 0.9145 |
| DIV2K validation | ×2 | 31.01 / 0.8923 | 33.66 / 0.9290 | 34.61 / 0.9372 | 34.87 / 0.9395 | 35.03 / 0.9407 | **35.07** / **0.9410** |
| | ×3 | 28.22 / 0.8124 | 30.09 / 0.8590 | 30.92 / 0.8734 | 31.10 / 0.8767 | 31.26 / 0.8795 | **31.30** / **0.8797** |
| | ×4 | 26.66 / 0.7512 | 28.17 / 0.8000 | 28.95 / 0.8178 | 29.10 / 0.8223 | 29.25 / 0.8261 | **29.29** / **0.8263** |
| FLOPs (M) | | - | 0.67 | 1.4 | 1.4 | 43 | 9.5 |

Table 5: Benchmark image denoising results of PSNR / SSIM for various noise levels. Training and testing protocols are followed as in [35]. The **best** results are in bold and the second are underlined.

| Dataset | Noise | BM3D | WNNM | DnCNN | Baseline | Sparse - | Sparse + |
|---|---|---|---|---|---|---|---|
| Set12 | 15 | 32.37 / 0.8952 | 32.70 / 0.8982 | 32.86 / 0.9031 | 32.97 / 0.9044 | 33.00 / 0.9048 | **33.04** / **0.9054** |
| | 25 | 39.97 / 0.8504 | 30.28 / 0.8557 | 30.44 / 0.8622 | 30.59 / 0.8655 | 30.63 / 0.8667 | **30.68** / **0.8676** |
| | 50 | 26.72 / 0.7676 | 27.05 / 0.7775 | 27.18 / 0.7829 | 27.40 / 0.7939 | 27.46 / 0.7954 | **27.51** / **0.7969** |
| BSD68 | 15 | 31.07 / 0.8717 | 31.37 / 0.8766 | 31.73 / 0.8907 | 31.79 / 0.8925 | 31.81 / 0.8928 | **31.83** / **0.8931** |
| | 25 | 28.57 / 0.8013 | 28.83 / 0.8087 | 29.23 / 0.8278 | 29.30 / 0.8311 | 29.33 / 0.8319 | **29.35** / **0.8327** |
| | 50 | 25.62 / 0.6864 | 25.87 / 0.6982 | 26.23 / 0.7189 | 26.35 / 0.7272 | 26.37 / 0.7265 | **26.39** / **0.7274** |
| Urban100 | 15 | 32.35 / 0.9220 | 32.97 / 0.9271 | 32.68 / 0.9255 | 32.94 / 0.9309 | 32.96 / 0.9316 | **33.05** / **0.9324** |
| | 25 | 29.70 / 0.8777 | 30.39 / 0.8885 | 29.97 / 0.8797 | 30.33 / 0.8930 | 30.36 / 0.8932 | **30.48** / **0.8959** |
| | 50 | 25.95 / 0.7791 | 26.83 / 0.8047 | 26.28 / 0.7874 | 26.76 / 0.8118 | 26.84 / 0.8113 | **26.95** / **0.8122** |
| FLOPs (M) | | - | - | 0.55 | 0.59 | 0.30 | 0.60 |

## 4.3 Main results

In this section, we compare our method on top of the state-of-the-art methods on image super-resolution, image denoising, and image compression artifact removal.

**Super-resolution.** We compare our method on top of EDSR [8], the state-of-the-art single image super-resolution methods, also with bicubic upsampling, VDSR [39]. As shown in Table 4, the small EDSR(S) has 16 residual blocks and 64 neurons per layer, and our sparse+ model extends it to 4 sparsity groups with cardinality 16 and outperforms on all benchmarks with 4× model capacity but negligible additional computation cost. The large EDSR(L) has 32 residual blocks and 256 neurons per layer, and our sparse- model has 32 residual blocks, 128 neurons per layer, 4 sparsity groups with cardinality 16. Then they have a similar model footprint and on-par benchmark accuracy but 4× computation cost difference.

**Denoising.** We compare our method with state-of-the-art image denoising methods: BM3D [40], WNNM [41] and DnCNN [35]. As shown in Table 5, our baseline model is residual networks with 16 blocks, 32 neurons per layer, 2× width multiplier [38, 42], and has similar footprint as DnCNN but better performance because of residual connections. Our sparse- model with 2 sparsity groups and 1× width multiplier keeps the model size as baseline but gains 2× computation reduction with better performance. Our sparse+ model adds 2 sparsity groups over baseline model, doubles model capacity, and boosts performance with negligible computation cost.

**Compression artifact removal.** We compare our method with state-of-the-art image compression artifact removal methods: JPEG, SA-DCT [43], ARCNN [44] and DnCNN [35]. As shown in Table 6,

Table 6: Compression artifacts reduction benchmark results of PSNR / SSIM for various compression qualities. Training and testing protocols are followed as in [35]. The **best** results are in bold.

| Dataset | $q$ | JPEG | SA-DCT | ARCNN | DnCNN | Baseline | Sparse - |
|---------|-----|------|--------|-------|-------|----------|----------|
| LIVE1 | 10 | 27.77 / 0.7905 | 28.65 / 0.8093 | 28.98 / 0.8217 | 29.19 / 0.8123 | 29.36 / 0.8179 | **29.39/0.8183** |
| | 20 | 30.07 / 0.8683 | 30.81 / 0.8781 | 31.29 / 0.8871 | 31.59 / 0.8802 | 31.73 / 0.8832 | **31.79/0.8839** |
| | 30 | 31.41 / 0.9000 | 32.08 / 0.9078 | 32.69 / 0.9166 | 32.98 / 0.9090 | 33.17 / 0.9116 | **33.21/0.9121** |
| | 40 | 32.35 / 0.9173 | 32.99 / 0.9240 | 33.63 / 0.9306 | 33.96 / 0.9247 | 34.18 / 0.9273 | **34.23/0.9276** |
| Classic5 | 10 | 27.82 / 0.7800 | 28.88 / 0.8071 | 29.04 / 0.8111 | 29.40 / 0.8026 | 29.54 / 0.8085 | **29.56/0.8087** |
| | 20 | 30.12 / 0.8541 | 30.92 / 0.8663 | 31.16 / 0.8694 | 31.63 / 0.8610 | 31.72 / 0.8634 | **31.72/0.8635** |
| | 30 | 31.48 / 0.8844 | 32.14 / 0.8914 | 32.52 / 0.8967 | 32.91 / 0.8861 | 33.07 / 0.8885 | **33.08/0.8891** |
| | 40 | 32.43 / 0.9011 | 33.00 / 0.9055 | 33.34 / 0.9101 | 33.77 / 0.9003 | 33.94 / 0.9028 | **33.96/0.9031** |
| FLOPs (M) | | - | - | 0.11 | 0.55 | 0.59 | 0.30 |

baseline and sparse- models have the same structure as the ones in denoising. Our method consistently saves computation and improves performance on all the benchmark datasets and different JPEG compression qualities.

## 5 Conclusions

In this paper, we have presented a method to structurally enforces sparsity constraints upon hidden neurons to achieve sparse representation in deep neural networks. Our method trade-offs between sparsity and differentiability, and is jointly learnable with deep networks iteratively. Our method is packed as a standalone module and substitutable for convolution layers in various models. Evaluation and visualization both illustrate the importance of sparsity in hidden representation for multiple image restoration tasks. The improved sparsity further enables optimization of model efficiency and capacity simultaneously.

## 6 Broader Impact

Image restoration algorithms can recover high-quality images from low-quality counterparts. The algorithms can help people who cannot afford professional cameras to take photos with low-end devices. However, many low-quality photos are taken under unwanted scenarios, *i.e.*, sneak shots. Powerful image restoration algorithms may contribute to related abuse. Low-level vision models cannot identify inappropriate images because of lacking the capability for high-level understanding of images. Our method can dramatically increase model capacity and increase the possibility to identify inappropriate patterns by models. Moreover, as shown in Fig. 6, our model can automatically explore high-level features, which benefits the capability to discover inappropriate images if we add corresponding supervision in training.

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
