[Supplementary Material]

# Supplementary Material for
# Neural Sparse Representation for Image Restoration

**Yuchen Fan, Jiahui Yu, Yiqun Mei, Yulun Zhang[†], Yun Fu[†], Ding Liu[‡], Thomas S. Huang**
University of Illinois at Urbana-Champaign, [†]Northeastern University, [‡]ByteDance
yuchenf4@illinois.edu

## 1 Proof of Equation 9

In Eq. 9, we reduce the soft sparsity constraints to the weighted sum of convolution kernels. The sparsity assumption plays an important role in the derivation, and its effects are also demonstrated in experiments with different softmax temperature settings in Table 1.

Here, we will give a detailed proof of the derivation process. We start from

$$Y = \sum_{i=1}^{k} \left( \beta_i W_2^i * F(W_1^i * X) \right).$$

because $F(\cdot)$ is assumed piece-wise linear, ReLU for example, we have $\lambda F(x) = F(\lambda x), \forall \lambda \geq 0$. Then

$$Y = \sum_{i=1}^{k} \left( \beta_i W_2^i * F(W_1^i * X) \right)$$
$$= \sum_{i=1}^{k} \left( \sqrt{\beta_i} W_2^i * F(\sqrt{\beta_i} W_1^i * X) \right).$$

The sparsity constraints are enforced here, where only one sparsity group is non-zero and the soft version relaxes it to only one sparsity group is weighted with one, and others are weighted with small values closed to zero. Formally, $\exists i$ denotes the index of the activated group, then

$$\text{s.t. } \beta_i \to 1 \text{ and } \beta_j \to 0, \forall j \neq i.$$

Hence,

$$F(\sqrt{\beta_i} W_1^i * X) \to F(\sum_{j=1}^{k} \left( \sqrt{\beta_j} W_1^j * X \right)) \text{ and } \sqrt{\beta_i} W_2^i * X' \to \sum_{j=1}^{k} \sqrt{\beta_j} W_2^j * X'.$$

Substitute into Y, then we have

$$Y = \sum_{i=1}^{k} \left( \sqrt{\beta_i} W_2^i * F(\sqrt{\beta_i} W_1^i * X) \right)$$
$$\approx \sum_{i=1}^{k} \left( \sqrt{\beta_i} W_2^i * F(\sum_{j=1}^{k} \left( \sqrt{\beta_j} W_1^j * X \right)) \right).$$

Figure 1: Unified network structure for image restoration (left). Residual block (right)

Grouped with common factors

$$Y \approx \sum_{i=1}^{k} \left( \sqrt{\beta_i} W_2^i * F(\sum_{j=1}^{k} \left( \sqrt{\beta_j} W_1^j * X \right)) \right)$$

$$= \left( \sum_{i=1}^{k} (\sqrt{\beta_i} W_2^i) \right) * F(\left( \sum_{i=1}^{k} (\sqrt{\beta_i} W_1^i) \right) * X).$$

## 2 Computation Complexity

In our paper, we claim the additional complexity of our method is negligible. Here we give a more detailed illustration of the comparison with conventional convolution.

Given input feature with spatial size $p \times p$ and channel number $c$, the conventional convolution with kernel size $q \times q$ and output channel number $c$ contains $q^2 c^2$ parameters in kernel (bias is ignored here). And the convolution operation needs $p^2 q^2 c^2$ multiply-add float operations.

In our method with $k$ sparsity groups, the number of parameters in kernels increases to $kq^2 c^2$. Additional computation in pooling is float $p^2 c$ operations. Given cardinality $d < c$, the number of additional parameters in MLP is $kcd < kc^2$, and the additional computations in MLP is $kcd < kc^2$. The weighted sum of convolution kernels also needs $kq^2 c^2$ multiply-add float operations. Compared with conventional convolution, the additional cost in our method is

$$\frac{p^2 c + kcd + kq^2 c^2}{p^2 q^2 c^2} < \frac{1}{q^2 c} + \frac{k}{p^2 q^2} + \frac{k}{p^2} \ll 1,$$

which is negligible, when $k \ll p^2$.

## 3 Network structure

In the experiments on image denoising, image compression artifacts removal, and ablation study, the baseline models are following the structure in [**? **]. As shown in Fig. 1, the structure is stacked by multiple residual blocks and additional convolution layers for input and output. The main branch is designed to learn corruption residual and add with a global skip connection directly from the input. The modules in blue color are only used for super-resolution task. The residual block has two convolution layers and an intermediate ReLU activation layer. The width multiplier denotes the ratio between the number of channels in ReLU activation and residual block inputs. We change both the convolution layers in residual blocks to our proposed method in experiments.

## 4 More Ablation Study

### 4.1 Pooling size

The pooling operation is applied to features before MLP to calculate averaging weights for convolution kernels. The window size is the only but crucial hyper-parameter in the pooling operation. When the

Figure 2: Ablation study on different pooling size.

Table 1: Ablation study on MLP depth.

| Depth | 1 | 2 |
|---|---|---|
| # of params (M) | 4.755 | 5.036 |
| FLOPs (M) | 1.2 | 1.2 |
| PSNR | 34.87 | 34.88 |

pooling size is too small, pooled features may not be stable and lead to an inaccurate combination of convolution kernels. When the pooling size is too big, pooled features may be over-smoothed and lead to an identical combination of convolution kernels for any input.

We conduct an ablation study with different sizes in pooling operation during training and test. Results in Fig. 2 show that: (1) the best pooling size for training is 64, smaller and bigger pooling size get worse results, which coincides with our analysis; (2) pooling size for evaluation needs to be smaller than the size for training to get better results.

## 4.2  MLP depth

We conduct an ablation study on the depth of MLP for averaging weights between sparsity groups. Results in Table 1 show that adding more layers and parameters in MLP improves the accuracy of averaging weights prediction and overall model performance, but the improvement is not significant. Hence, models in our experiments come with only one layer in MLP.

## 5  Implementation

We implement our method in PyTorch. Our method inherit the conventional convolution layer with two additional hyper-parameters *num_sparsity_groups* and *num_sparsity_cardinals* to specify the number of sparsity groups and cardinality respectively. Hence our method can be used to replace conventional convolution layers in any model.

Here is the code snippet (based on PyTorch >= 1.5):

```
class Conv2d(nn.Conv2d):

    def __init__(self,
                 in_channels,
```

```python
                out_channels,
                kernel_size,
                num_sparsity_groups=1,
                num_sparsity_cardinals=1,
                stride=1,
                padding=0,
                dilation=1,
                groups=1,
                bias=True,
                padding_mode='zeros'):
    self.num_sparsity_groups = num_sparsity_groups
    self.num_sparsity_cardinals = num_sparsity_cardinals
    out_channels *= num_sparsity_groups
    super(Conv2d, self).__init__(in_channels, out_channels,
    kernel_size, stride,
                                    padding, dilation, groups, bias,
    padding_mode)
    self.mlp = nn.Sequential(
        nn.Linear(in_channels,
                self.num_sparsity_groups * self.
    num_sparsity_cardinals),)
    if self.padding_mode == 'circular':
        raise NotImplementedError

def _conv_forward(self, input, weight):
    sparsity = self.mlp(input.mean([-1, -2]))
    sparsity = sparsity.view(
        (-1, self.num_sparsity_groups, self.num_sparsity_cardinals))
    sparsity = nn.functional.softmax(sparsity, dim=1)
    weight = weight.view((self.num_sparsity_groups, self.
    num_sparsity_cardinals,
                        -1, *weight.shape[1:]))
    weight = torch.einsum("abc,bcdefg->acdefg", (sparsity, weight))
    weight = weight.reshape((-1, *weight.shape[3:]))
    bias = self.bias.view(self.num_sparsity_groups, self.
    num_sparsity_cardinals,
                        -1)
    bias = torch.einsum("abc,bcd->acd", (sparsity, bias))
    bias = bias.reshape(-1)
    batch_size = input.shape[0]
    input = input.view((1, -1, *input.shape[2:]))
    output = nn.functional.conv2d(input, weight, bias, self.stride,
                                    self.padding, self.dilation,
                                    self.groups * batch_size)
    output = output.view((batch_size, -1, *output.shape[2:]))
    return output
```

# 6   Visualization

## 6.1   Super-resolution

We further visually compare our MAN with other state-of-the-art approaches. As shown in the Figure 3, our method produces higher quality images than others: SRCNN [1], FSRCNN [2], VDSR [3], LapSRN [4], MemNet [5], EDSR [6] and SRMDNF [7].

### 6.1.1   Denoising

More visual results on Set12 dataset are shown in Figure 4.

### 6.1.2   Compression artifact removal

More visual results on LIVE1 dataset are shown in Figure 5.

| HR | Bicubic | SRCNN | FSRCNN | VDSR |
| LapSRN | MemNet | EDSR | SRMDNF | Ours |

img_004

| HR | Bicubic | SRCNN | FSRCNN | VDSR |
| LapSRN | MemNet | EDSR | SRMDNF | Ours |

img_020

| HR | Bicubic | SRCNN | FSRCNN | VDSR |
| LapSRN | MemNet | EDSR | SRMDNF | Ours |

img_073

Figure 3: Visual comparison results of $\times 4$ image super-resolution on Urban100 dataset

Figure 4: Visualization for image denosing.

Figure 5: Visualization for JPEG compression artifact removal.