[Reviews · NeurIPS 2020]

Review 1

Summary and Contributions: The paper focuses on sparse representation in deep neural networks for image restoration tasks. Authors propose adding sparse constrains via model structure instead of common regularization in the loss function. The method determines a subset of convolution kernels dynamically during inference and saves computation of the complementary set of convolution kernels. The method gives a better tradeoff between speed and accuracy, and it can be plug-in any convolution neural networks. 

Strengths: 1. Novelty and importance: the paper addresses one of the most important concepts of sparse representation which dominates the ill-posed image restoration tasks for decades and is rarely investigated after deep learning approaches prevalent. 2. Methodology: the paper borrows ideas from sparse coding based approaches but implements them in a deep learning favorable way. The sparsity is determined by an MLP conditioning on features instead of interactive optimization in sparse coding. 3. Experiments: ablation study covers the key design choices of the proposed method. The main results show significant improvement in terms of speed and accuracy over various models and tasks. 4. Potential: the proposed solution significantly outperforms CondConv on image restoration, so it can be extended to image recognition tasks. More importantly, this work gives a promising direction to explain and improve CondConv theoretically.

Weaknesses: 1. The experiment mentioned Line 140 is not addressed in the ablation study. 2. Results in Figure 4 shows that increasing cardinality can improve performance constantly. Is there any regression point?

Correctness: The paper is technically correct.

Clarity: The paper is written concisely and clearly. Technically details are enriched in the main paper. 

Relation to Prior Work: Related works are well addressed. Comparisons between proposed and previous methods are covered both in method and experiment sections. Covered previous works are not limited to EDSR and RCAN in image restoration, but also SE-Net and CondConv in image recognition.

Reproducibility: No

Additional Feedback:


Review 2

Summary and Contributions: Authors introduce a form of sparsity for deep CNNs, by which a subset of channels are turned on/off by auxilliary soft gating networks (each of which is a multi-layer perceptron on the input signal). They apply the network to super-resolution, denoising, and compression artifact removal, showing performance at state-of-the-art for all three.

Strengths: Results are at state-of-the art, with about half the computation (at least for the super-resolution application).

Weaknesses: * Model is ad-hoc, somewhat complex, and difficult to understand (as described). * Interpretation/Visualization of gating network (fig 5) was not so helpful. The authors state that "the last layer is more correlated to semantics, for example, tree branches in the first image and lion in the second image." But I do not see this at all. Moreover, I don't see why would one expect a semantic partition for a network that is performing super-reoslution?

Correctness: Method is ad-hoc, so no derivations/proofs to check. I can't verify consistency of primary equations without understanding definitions of c or d (see below).

Clarity: I had trouble understanding the details of the architecture. Figure 3 did not help. In particular, after multiple re-reeadings I still could not understand the definitions of the hidden nodes c or cardinal groups d. Also, please have your article proof-read for style and grammar issues - there are many improper sentences.

Relation to Prior Work: The method is an extension of the Squeeze-and-Excitation model (CVPR-2018), and offers some advantages over that.

Reproducibility: No

Additional Feedback: >>> Added after reading author's feedback, and other reviewer comments: The use of gating networks in these problems is interesting, and gives performance gains. I still think the paper is not very clearly written, but acknowledge that the other reviewers did not have this reaction. The author feedback helped me to understand fig5, but I still would not call that "semantic" - there is nothing that indicates the meaning or idenity of the scene or objects, just foreground vs. background. I agree with R3 that the connection to group sparsity should be explained, and with R4 that connections to attention models should be explained. Overall, I did not find the feedback or other reviews significantly altered my view of the paper, which is marginally above threshold. In any case, I hope the authors will find our comments helpful in improving the work.


Review 3

Summary and Contributions: This paper enforces sparsity in neural networks in order to improve the network performances. The authors argue that iterative sparse coding algorithms are not appropriate because of their iterative nature. They observe that ReLU impose a sparsity. The paper proposes novel sparsity constraints to obtain a sparse representation in neural networks. Their algorithm enforces a group sparsity property. They do it with maximum selection operation similarly to a matching pursuit algorithm, To get a differentiable operator they use a softmax. In their experimental section the authors did a serious ablation study testing different hypoetheses including sparsity constraints. They have tested it on a wide range of applications including noise removal, compression and super-resolution.

Strengths: This paper did a serious experimental work to test ideas to impose sparsity in neural networks with different sparsity criteria. The expriencs have been carried with care.

Weaknesses: The paper is rather weak on the theoretical side of sparsity and the existing work. The paper claims in the introduction that "sparsity of hidden representation in deep neural networks cannot be solved by iterative optimization as sparse coding". I do not understand this claims since algorithms such as LISTA do compute sparse coding from few layers in deep networks. The fact that sparsity is needed to do denoising, compression or inverse problems is well understood independantly from neural networks and result from work carried by may researchers such as Donoho between 1995 and 2005. I do n \ot understand why they say that such sparsity can not be implemented given that a ReLU is the proximal operator of a positive l1 sparse coder, that many algorithms implement a sparse code with such architectures, and that such architectures with ReLU get very good performance for denoising and inverse problems as shown by "Convolutional Neural Networks for Inverse Problems in Imaging: A Review" published in 2017, and much more work has been done so far. Doing group sparse coding is not either a problem by using mixed l2/l1 norms which are also implemented with ReLU non-linearities. The algorithm proposed by the authors makes sense and in fact has similarities with matching pursuit procedure which extract maxima as they do to build sparse codes, but there is no guarantee of efficiency of such algorithms in this neural network context, beside the numerical experiments shown at the end. The difficulty to evaluate such experiments is that there are often marginal improvement relatively to methods which are not always state of the art such as JPEG or SA-DCT for compression. The multiplication of tables and numbers do not help.

Correctness: The empirical methodology seems correct but some of the claims of the paper concerning sparsity are not correct.

Clarity: It is sufficiently well written to be understood.

Relation to Prior Work: Yes, this is an algorithmic work and this particular algorithm has not been tested so far.

Reproducibility: Yes

Additional Feedback:


Review 4

Summary and Contributions: This paper presents a new approach to enforce sparsity in neural network based denoising. The method is well explained, seems novel and promising and results are convincing.

Strengths: The denoising results are convincing, the paper is well written, figures and tables are neat, the method is original.

Weaknesses: It is missing comparisons in terms of computation time. Minor comments: Line 62: What is coupled dictionary? Line 172: How does the model behaves is the noise is lower than 15 or higher than 50. Line 168: Use "$\times 2$" instead of "x2"

Correctness: Yes it seems correct to me.

Clarity: Yes, it is well written.

Relation to Prior Work: Yes this paper is very clear.

Reproducibility: Yes

Additional Feedback: Adding computation time in addition to PSNR and SSIM would be relevant. Comparing with a patch based approach such as FEPLL for denosing and super resolution would be useful too: Parameswaran, Shibin, et al. "Accelerating GMM-Based Patch Priors for Image Restoration: Three Ingredients for a $100 {\times} $ Speed-Up." IEEE Transactions on Image Processing 28.2 (2018): 687-698.

[Author Response · NeurIPS 2020]

We thank all the reviewers. Our novelty is appreciated by R1, R3, and R4, the superior performance is recognized by R1, R2, and R4, and the writing clarity is accredited by R1, R3, and R4. Below we addressed all questions and concerns.

**R1: shared $\beta$.** Following the settings in Table 1, the model with shared $\beta$ drops PSNR from 34.87 dB to 34.85 dB.

**R1: cardinality & performance.** Increasing cardinality can constantly improve performance when model is under-parameterized. Since our ablation study is based on small models, the cardinality is limited by the number of channels. We do observe performance regression point when reducing the size of training dataset.

**R2: cardinality.** The most primitive formulation of cardinality groups is that we start from single convolution layer with only $c/d$ (/ denotes numeric division) output nodes and build sparse representation with $k$ groups upon it as Section 3.2, then each sparsity group has $c/(dk)$ nodes and the MLP has $k$ outputs. The cardinality groups repeat building this structure for $d$ times and concatenate output nodes of all the $d$ convolutions. In section 3.3, we move the concatenation operation from convolution output nodes to convolution kernels for formulation simplification.

**R2: visualization.** Dark blue and light yellow reflect the most certain regions of sparsity group selection. In the second column of Fig. 5, dark blue more likely co-occurs with brown and black texture in original RGB images and light yellow co-occurs with green textures. In the last column, dark blue and light yellow occur only within the region of tree and sky, lion, and squirrel, and especially concentrate in foreground in the second and third images. We think it is more related to semantic, since foreground can be automatically identified and receptive field grows as stacking convolution layers. We will remove the statement about semantic if you still think it is not conclusive.

**R3: sparse coding & LISTA.** We focus on exploring sparsity of hidden representation in deep neural networks instead of implanting deep structures in sparse coding approaches, such as LISTA. Although LISTA variants could be unrolled into similar formulation as deep models, the iterative shrinkage algorithms solve sparse coding with several folds of computation cost than a single convolution layer, which is hard to be applied in efficient deep models without parameter sharing across layers. In contrast, our method improves the efficiency of deep models, meanwhile increases the number of distinct parameters. We will revise the statement in line 33 as "cannot be efficiently solved" to reduce ambiguity.

**R3: sparsity by ReLU.** As we discussed in Section 2.3, sparsity by ReLU has been addressed in [10]. ReLU thresholds negatives to zeros and has a gap with sparsity definition in Eq. 1, where the number of non-zero values should be much less than representation dimensionality. Also, ReLU can only determine sparsity for given vectors, but our method predicts the sparsity by a side MLP and saves the computation for unnecessary elements. Experiments show our method constantly outperforms networks only with ReLU activations.

**R3: group sparse coding.** Our method with multi-level sparsity constraints by ReLU and Softmax shares a similar idea with mixed L2 and L1 norms in group sparse coding. Furthermore, enforcing the constraints into model structure instead of additional objective functions is more favorable for efficiency and performance in deep models. We will add more detailed connection and comparison between group sparse coding and our method in the future version.

**R3: guarantee of efficiency.** First, our method constantly outperforms baselines over multiple benchmarks including hundreds of images with either less computation and the same model size or the same computation and larger model size. Second, there are many design choices in our methods, and results in the ablation studies coincide with our design and analysis. Third, visualization shows non-trivial sparsity group selection.

**R3&R2: state-of-the-art baselines.** We focus on the importance of sparse representation of deep models and propose a very general approach to achieve sparsity within individual convolution layer, instead of a single state-of-the-art model. Hence, all the baselines are residual networks with only convolutional and activation layers. Other advanced techniques, such as multi-scale (U-net) and self-attention (non-local), are perpendicular to our method and can be easily combined with ours for superior performance.

**R4: coupled dictionary.** The coupled dictionary is introduced in [3], where encoding dictionary $D_1$ and decoding dictionary $D_2$ are independent two dictionaries. In previous method [1], $D_1$ and $D_2$ are the same one.

**R4: computation time.** We follow the protocol as Squeeze-and-Excitation networks using CPU inference time to address the latency for embedded device applications. We evaluate the forward running time of a single convolution layer of our implementation in PyTorch. The running time is the average of

Computation time (in milliseconds).

| Group size | N/A | 2 | 4 | 8 | 16 |
|---|---|---|---|---|---|
| fixed model size | 1.96 | 1.17 | 0.77 | 0.56 | 0.70 |
| fixed FLOPs | 1.96 | 2.12 | 2.12 | 2.08 | 2.08 |

100 times on quad-core Intel Core i5-2500 at 3.30GHz. We set batch size as 100, input and output channels as 64, patch size as 64. The time for additional computation is around 5-10%. Our implementation can be improved, since SE-Net takes 3ms over 164ms and our approach is more efficient by only modifying convolution kernels instead of features.

**R4: patch-based approach.** We will add the suggested paper and more methods with fair comparisons including patch-based deep models, such as channel attention and self-attention models.

[Meta-Review · NeurIPS 2020]

The paper initially received mixed ratings. After reading the rebuttal, all reviewers agree that the paper should be accepted. Yet, a few concerns have been raised, which should be taken into account for the final version of the paper.